# Zinc Oxide and Copper Chitosan Composite Films with Antimicrobial Activity

**DOI:** 10.3390/polym13223861

**Published:** 2021-11-09

**Authors:** Candy del Carmen Gamboa-Solana, Martha Gabriela Chuc-Gamboa, Fernando Javier Aguilar-Pérez, Juan Valerio Cauich-Rodríguez, Rossana Faride Vargas-Coronado, David Alejandro Aguilar-Pérez, José Rubén Herrera-Atoche, Neith Pacheco

**Affiliations:** 1Facultad de Odontología, Universidad Autónoma de Yucatán, Calle 61 A #492 A x 90 y Av. Itzáes, Centro., Mérida C.P. 97000, Mexico; a18018519@alumnos.uady.mx (C.d.C.G.-S.); fernando.aguilar@correo.uady.mx (F.J.A.-P.); david.aguilar@correo.uady.mx (D.A.A.-P.); jose.herrera@correo.uady.mx (J.R.H.-A.); 2Centro de Investigación Científica de Yucatán, Unidad de Materiales, Calle 43 No. 130 x 32 y 34, Colonia Chuburná de Hidalgo, Mérida C.P. 97205, Mexico; jvcr@cicy.mx (J.V.C.-R.); ross@cicy.mx (R.F.V.-C.); 3Centro de Investigación y Asistencia en Tecnología y Diseño del Estado de Jalisco, A.C. Parque Científico Tecnológico de Yucatán, km 5.5 Carretera, Sierra Papacal-Chuburná, Chuburná C.P. 97302, Mexico; npacheco@ciatej.mx

**Keywords:** chitosan, glutaraldehyde, zinc oxide, copper nanoparticles, *S. aureus*, *S. typhimurium*

## Abstract

The role of the oral microbiome and its effect on dental diseases is gaining interest. Therefore, it has been sought to decrease the bacterial load to fight oral cavity diseases. In this study, composite materials based on chitosan, chitosan crosslinked with glutaraldehyde, chitosan with zinc oxide particles, and chitosan with copper nanoparticles were prepared in the form of thin films, to evaluate a new alternative with a more significant impact on the oral cavity bacteria. The chemical structures and physical properties of the films were characterized using by Fourier transform infrared spectroscopy (FTIR,) Raman spectroscopy, X-ray photoelectron spectroscopy (XPS), elemental analysis (EDX), thermogravimetric analysis (TGA), X-ray diffraction (XRD), scanning electron microscopy (SEM), and contact angle measurements. Subsequently, the antimicrobial activity of each material was evaluated by agar diffusion tests. No differences were found in the hydrophilicity of the films with the incorporation of ZnO or copper particles. Antimicrobial activity was found against *S. aureus* in the chitosan film crosslinked with glutaraldehyde, but not in the other compositions. In contrast antimicrobial activity against *S. typhimurium* was found in all films. Based on the data of present investigation, chitosan composite films could be an option for the control of microorganisms with potential applications in various fields, such as medical and food industry.

## 1. Introduction

One of the most complex and heterogeneous microbial communities in the human body is found in the oral cavity [1]. Biofilm bacteria play an essential role in the origin of oral diseases, including dental caries, gingivitis, and periodontitis. Despite extensive research, the ideal antimicrobial for use in the oral cavity has not yet been found. One of the bacteria found in the oral cavity, according to several studies, is *Staphylococcus* spp. This Gram-positive bacterium lodges in healthy patients’ nostrils, ears, and oral cavity [2,3,4]. *S. aureus* can be responsible for infections associated with oral and dental health care. Patients with high levels of *S. aureus* in saliva are potential sources of infection in a dental office, as it is spread in the environment during dental therapy [2]. In addition, *S. aureus* is responsible for a wide variety of conditions that can cause infections with high mortality rates. Alternatives to antibacterial agents have been sought due to their resistant strains; one of these alternatives is chitosan [5].

Chitosan (CHT) is produced from chitin, which is a natural polysaccharide found in the exoskeleton of crabs, shrimp, lobsters, corals, squid, jellyfish, as well as insects, fungi, yeasts, and algae, and is the second most abundant natural polymer after cellulose [6,7,8]. The solubility of chitosan provides opportunities to manufacture it in many ways [9]. Among its characteristics are biodegradability, biocompatibility, and antimicrobial properties, since it inhibits the growth of a wide variety of bacteria [6,10].

The characteristics of chitosan depend on structural parameters such as molecular weight and its degree of deacetylation. The degree of deacetylation strongly influences the physical, chemical, and biological properties [11]. In addition, the source of extraction and the procedures adapted to carry out the deacetylation can affect the final properties.

One of the main reasons for chitosan possessing antimicrobial activity is a positively charged amino group at pH values below 6.3 (carbon 2), which interacts with negative charges of the cell wall of microorganisms. That interaction generates a breakdown or lysis of these structures, leading to losing protein compounds and other intracellular constituents of bacteria [12].

Glutaraldehyde (GA) has been used to improve the properties of chitosan [13]. It has been reported that the addition of glutaraldehyde to chitosan films contributes to antimicrobial activity [14]. Glutaraldehyde produces Schiff bases between aldehyde groups and free amine groups in deacetylated chitosan polymers. Thus, GA improves the mechanical properties, fixes the structure, and modifies permeation of chitosan [15]. 

In recent years, chitosan coating with metal oxide nanoparticles has also received attention, particularly zinc oxide nanoparticles, due to their less toxic, environmentally friendly, and diverse applications [16]. Various antibacterial agents, such as copper nanoparticles, have also been combined with chitosan to enhance their antimicrobial activity [17]. The choice of chitosan as a stabilizer for copper nanoparticles is due to its ability to chelate metals, which makes it a perfect candidate for the synthesis of metal nanoparticles [18]. Coupling chitosan with metal nanoparticles can maximize their antibacterial and antifungal potential [12].

The objective of this study is to evaluate the physicochemical characteristics and antimicrobial activity of chitosan-based composites thin films prepared with, glutaraldehyde (GA), zinc oxide (ZnO) and copper nanoparticles. At the same time, to provide some evidence about the synergy of the components, ZnO, Cu, and chitosan, arranged in the same material, and determining if their antimicrobial capacity improves when used together.

## 2. Materials and Methods

### 2.1. Materials

Chitosan (molecular weight 223.332 g/mol and with 70–80% deacetylation degree), Acetic Acid, Sodium Hydroxide, Glutaraldehyde grade II at 25% (molar mass 100.11 g/mol) and zinc oxide (ZnO) (powder, ASC reagent, CAS 1314-13-2, SKU 205532) reagents were acquired from Sigma-Aldrich (Saint Louis, MO, USA). Copper nanoparticles (nCu) in aqueous suspension were acquired from the manufacturer Nano Process SPA (Antofagasta, Chile). The determination of nCu particle size was performed by dynamic light scattering (DLS) using a NANOTRAC WAVE II ©Microtrac Retsch GmbH (York, PA, USA), following the methodology described elsewhere [19]; the average particle size was 27.4 nm. For ZnO particle size was 55 μm, measured by LS100 Coulter Particle Size Analyzer from Beckman Coulter, Inc (Indianapolis, IN, USA).

### 2.2. Preparation of Chitosan Films

#### 2.2.1. Preparation of Unfilled Chitosan Films

Initially, a solution with 200 mg of chitosan (CHT) was prepared by dissolving it in 30 mL of acetic acid at 0.4 M and a pH of 4.5. The solution was stirred for one hour until complete dissolution, then poured into plastic Petri dishes and dried at 25 °C for approximately five days until complete evaporation of the acetic acid. The resulting films were neutralized with 5% sodium hydroxide (NaOH) solution and washed with distilled water. Finally, the films were left to dry at room temperature (approx. 25 °C) for approximately two days. 

#### 2.2.2. Preparation of Glutaraldehyde Crosslinked Chitosan Films

Crosslinked GA-chitosan films were obtained, and 200 mg of chitosan was dissolved in 30 mL of acetic acid. The solution was stirred in a magnetic stirrer plate at 25 °C for one hour. After the dissolution of chitosan, 0.3744 mM (150 µL) of a 25% glutaraldehyde (GA) was added. The mixture was left under magnetic stirring for 5 h until complete homogenization of the mixture and then the solution was poured into plastic Petri dishes and dried at 25 °C. Films obtained after solvent casting were neutralized with 5 wt.% NaOH and rinsed with distilled water. Finally, films dried at 25° for 2 days.

#### 2.2.3. Preparation of Composite Chitosan Films

After dissolving the chitosan with acetic acid and stirring for one hour, the proper amount of each antibacterial particle was added, for each different composite (Table 1). The solution was left under magnetic stirring for 1 h and sonicated for 15 min in an ultrasonic bath. Finally, the solution was kept in magnetic stirring for 2 min. The solutions were poured into plastic Petri dish, dried at RT (25 °C) for evaporation of the residual acetic acid, and neutralized with 5% NaOH. Finally, they were washed with distilled water and left to dry at RT for seven days.

### 2.3. Composition and Structural Characterization of CHT Composite Films

#### 2.3.1. Fourier Transform Infrared Spectroscopy (FTIR)

Fourier transformed infrared spectroscopy (FTIR)-spectra were obtained from chitosan and modified chitosan films using a Nicolet Thermo-Scientific 8700 spectrophotometer (Madison, WI, USA) using attenuated total reflectance (ATR) technique. The spectra were obtained in the range of 4000 to 600 cm^−1^, with a Zinc selenide crystal, averaging 100 scans with a resolution of 4 cm^−1^, and correction for H_2_O and CO_2_.

#### 2.3.2. Raman Spectroscopy

Raman spectra were obtained using the InVia™Raman Renishaw microscope (Wottonunder-Edge, Gloucestershire, UK). A 633 nm argon laser at a power of 50%, was used as the excitation radiation source. The samples were analyzed in the spectral range of 3200 to 100 cm^−1^ with 2 accumulations, 1800 grid, 50× objective, with an exposure time of 10 s.

#### 2.3.3. Thermogravimetric Analysis (TGA)

The thermal characterization of chitosan and modified chitosan films was carried out by TGA with a Perkin Elmer TGA-7 (Waltham, MA, USA), in a temperature range of 45 °C to 700 °C, at a heating rate of 10 °C/min, under a nitrogen atmosphere.

#### 2.3.4. X-ray Diffraction (XRD)

An X-ray diffraction study was carried out to know the internal structure of the chitosan composites in a Bruker D-8 Advance diffractometer (Karlsruhe, Germany), operating with a Cu Kα radiation at a wavelength of 1.54 Å, in a 2θ range of 10° to 60°, at a step count of 5 s, passage time of 0.02° at 40 kV and 30 mA.

### 2.4. Surface Properties of CHT Composite Films

#### 2.4.1. Scanning Electron Microscopy (SEM)

The morphology of films surface was observed in a JEOL, JMS 6360LV (Akishima, Tokyo, Japan) with an accelerating voltage of 20 keV. In addition, Energy-dispersive X-ray spectroscopy (EDX) (Oxford Instruments, INCA X-Sight 7582, High Wycombe, UK) coupled with the microscope, was used to obtained elemental surface composition. At least three different locations (top, middle, and bottom) were scanned, and the average reported. The samples (1 cm diameter) were previously plated with gold on a DESK II Denton Vacuum metallizer Coater (Moorestown, NJ, USA) during 50.0 s, with an accelerating voltage of 8.00 kV and an energy of 1.3 eV.

#### 2.4.2. X-ray Photoelectron Spectroscopy (XPS)

A general inspection spectrum survey from 0 to 1200 eV was obtained to identify the chemical elements present in each sample. Using a Thermo Scientific K-Alpha X-ray photoelectron spectrometer (Waltham, MA, USA), (with a monochromatic source of Al Kα with an energy of 1486.6 eV).

#### 2.4.3. Contact Angles

The contact angle measurement was performed using a ramé-hart model 250 goniometer/tensiometer with DROPimage Advanced v2.8 (Succasunna, NJ, USA), with the sessile drop technique (5 µL), using distilled water, phosphate-buffered saline (PBS), and Dulbecco’s Modified Eagle’s Medium (DMEM). Measurements recorded at 25 °C. Three replicates per sample were averaged, the image of the water or DMEM droplet was captured within 10 s of delivery. The contact angle was measured automatically using computer integrated software.

### 2.5. Biological Studies

#### Antimicrobial Activity Assays

An agar diffusion assay was conducted to determine antibacterial activity against *Staphylococcus aureus* ATCC 25923 and *Salmonella typhimurium* ATCC 14028 strains. Both bacteria strains were cultured in nutrient broth for reactivation at 37 °C for 24 h. Subsequently, strains were inoculated on Mueller–Hinton agar prepared in Petri dishes using 100 µL of the suspension of each bacterium separately at a concentration adjusted with a saline solution of 106 CFU/mL; the microorganisms were diffused in the medium using a glass loop. Later, disks of 6 mm in diameter of the chitosan and modified chitosan films were cut and placed in the Petri dishes containing the Mueller–Hinton culture medium previously inoculated with the microorganisms, and then were incubated at 37 °C for 24 h.

Antimicrobial activity was determined as negative or positive, according to the presence of microorganism growth above or below the film disks, when observed by a wireless digital microscope SKYBASIC Digital Microscopes, model XWJ303, (San Francisco, CA, USA), with 500× magnification and 1920 X 080p resolution. *P*-iodonitrotetrazolium chloride solution (10 μL at 0.2 mg/mL) was used as the growth indicator to identify the presence or absence of bacterial growth; the presence of red-pink color change was verified after 5 min of incubation.

## 3. Results and Discussion

### 3.1. Physicochemical and Structural Characterization of Modified Chitosan Films

#### 3.1.1. FTIR Spectroscopy

FTIR spectra of pristine chitosan (CHT) and mixtures of chitosan/glutaraldehyde (CHT-GA), chitosan/zinc oxide (CHT-ZnO), chitosan/copper nanoparticles (CHT-Cu), and chitosan/zinc oxide/copper nanoparticles (CHT-ZnO-Cu) are shown in Figure 1.

Pristine chitosan showed strong absorption band at 3266 cm^−1^ due to the stretching vibration of the O–H and N–H bonds. The bands at 2925 cm^−1^ and 2879 cm^−1^ were associated to methyl groups [19,20]. Chitosan showed a characteristic band of amide-I (1637 cm^−1^) derived from the non-deacetylated residues of chitosan [21], amide II (1548 cm^−1^), and amide III (1323 cm^−1^): this band is commonly used to calculate the degree of acetylation. The bands found at 1412 cm^−1^ and 1374 cm^−1^ represent the deformation modes of the C–H_2_ and C–H_3_ bonds. Bands between 1154 cm^−1^ and 1027 cm^−1^ are assigned to C–O–C of the glycosidic bonds. The bands observed in the 1155 cm^−1^ and 800 cm^−1^ are known to be C–O stretch vibrations of chitosan [22]. 

The FTIR spectra of CHT-glutaraldehyde film showed the increase of the band corresponding to the amide-I, while shifting to higher wavenumbers up to 1648 cm^−1^. When chitosan was crosslinked with glutaraldehyde, the band at 1648 cm^−1^ is attributed to amide I and imine bond [23]. In contrast, the amide-II decreased and was displaced toward a higher wavenumbers or higher frequency (1570 cm^−1^). Another interesting fact is the disappearance of the band located at 1154 cm^−1^, which coincides with that reported by other authors [24]. In this regard, Chen et al. compared the spectra of chitosan, chitosan with silicotungstic acid hydrate (HAS) and chitosan with HAS crosslinked with GA. They observed that the intensity of the band of hydroxyl groups for chitosan with silicotungstic acid hydrate was slightly more intense than in the sample crosslinked with GA: this would indicate that GA made the sample less hydrophilic [25].

Chitosan film with 5% of Zn-O, presented a slight change, in the bands of 3294 cm^−1^, attributed to the O–H stretching mode of the hydroxyl group, 2877 cm^−1^. Amide I and amide II appeared at 1648 cm^−1^ and 1574 cm^−1^ while the intense absorption at 1423 cm^−1^ reduced its intensity. Peaks at 1374 cm^−1^ and 1315 cm^−1^ were clearer, the former being of higher intensity. Bands at 1151 cm^−1^, 1030 cm^−1^, and 896 cm^−1^ remained, although the first one can be also assigned to Zn-O bonds. Additionally, we observed in pristine ZnO bands at 987, 875, 781, 730, and 660 cm^−1^ (Appendix A) which confirms the presence of zinc oxide particles as reported elsewhere [26,27]. Dobruka and Dugaszweska reported that the bands attributed to the vibration of the Zn-O–Zinc bond are found at 400 cm^−1^ to 600 cm^−1^ [28] while Abdolhossien et al. also found a band at 438 cm^−1^ which confirmed the presence of the zinc oxide particles [29]. In a study by Yousseff et al. when comparing the pure chitosan film with the chitosan film with ZnO, they found new absorption bands at 500 cm^−1^ and 580 cm^−1^, that were attributed to the vibration of the O-Zn-O groups respectively [30]. 

In the FTIR spectrum of the copper nanoparticles (Appendix A) bands at 2918 cm^−1^, 1557 cm^−1^, 1291 cm^−1^, 994 cm^−1^, 856 cm^−1^, 758 cm^−1^, 666 cm^−1^, 638 cm^−1^, 628 cm^−1^, and 608 cm^−1^ were observed. Some studies report that bands recorded from 700 cm^−1^ to 400 cm^−1^ are assigned to Cu–O vibrations, which would confirm the presence of Cu_2_O [31] whereas Ceja-Romero et al. reported Cu–O bands at 572 cm^−1^ [32]. Despite the low copper concentration used, our study, a shoulder at 947 cm^−1^, and bands at 895 cm^−1^, 875 cm^−1^, 715 cm^−1^ and 659 cm^−1^ were observed that coincides with the chitosan and Cu film (Figure 1). Maldonado et al. observed that the presence of copper nanoparticles in the chitosan matrix produces a decrease in the intensity of the signal associated with the vibrations of the hydroxyl and amino groups (O–H/N–H) of the polymer chain, which indicates that it is proportional to the concentration of added particles. This suggests that the copper nanoparticles and the Cu^2+^ ions released during mixing in solution interact with the H-O/N-H groups of the chitosan chain, allowing the formation of chelates [33]. Gritsch et al. compared chitosan and chitosan samples with Cu and found a decrease in the amide and amine bands at 1650 cm^−1^ and 1600 cm^−1^ [34].

Regarding the chitosan films with Zn-O and Cu particles, we found differences in the spectra as the amide II was observed at 1577 cm^−1^ and the peaks at 1466 cm^−1^ and 1420 cm^−1^ decreased in intensity when the copper and zinc oxide particles are added, compared to pristine chitosan film (Figure 1). It is observed that the band at 1648 cm^−1^ appear in ZnO/chitosan film slightly shifted to a 1645 cm^−1^ in chitosan with Zn-O and Cu film. This band was also found by Miri et al., who attribute it to the O–H stretching mode of the hydroxyl group [29]. In addition, bands at 821 cm^−1^, 757 cm^−1^, 695 cm^−1^ and 664 cm^−1^ were noted in the composite.

The calculation of the *N*-acetylation of chitosan was carried out following the Brugnerotto formula, taking the characteristic bands observed in (Appendix A) of the amide III located at 1323 cm^−1^ and as a reference the band of the secondary amide at 1407 cm^−1^ [35]. Finally, a deacetylation degree of 88.66% was obtained which is closed to the manufacturer reported value.

#### 3.1.2. Raman Spectroscopy

Figure 2 shows Raman spectra of chitosan (CHT) and chitosan mixed with GA (CHT/GA), zinc oxide (CHT/Zn-O), copper nanoparticles (CHT/Cu), and zinc oxide/copper nanoparticles (CHT/Zn-O/Cu).

The main signals showed in chitosan films were at 2930 cm^−1^ and 2917 cm^−1^ attributed to the stretching vibration *ѵ*(CH_2_); 1604 cm^−1^ assigned to the bending vibration in the δ(NH_2_) plane; 1382 cm^−1^ assigned to combined in plane bending vibrations of various groups δ(CH_2_), δ(CH), δ(OH); and at 1093 cm^−1^ attributed to combined vibrations of groups *ѵ*(C–O–C) + *ѵ*(ϕ) + *ѵ*(C–HO) + *ѵ*(C–CH_2_) + δ(CH) − ρ(CH_2_) + ρ(CH_3_). At 925 cm^−1^ and 898 cm^−1^ medium intensity absorption were also detected.

Chitosan films crosslinked with glutaraldehyde exhibited a strong absorption at 638 cm^−1^ with a small shoulder at 573 cm^−1^. The presence of 5 wt.% of ZnO in CHT films enhanced absorptions at 1374 cm^−1^, 1107 cm^−1^, and 893 cm^−1^. In addition, we observed absorptions at 208 cm^−1^, 320 cm^−1^, 424 cm^−1^, and 622 cm^−1^. Miri et al. found a region of high intensity between 300 cm^−1^ and 600 cm^−1^ and a region of low intensity between 100 cm^−1^ and 120 cm^−1^ in the Raman spectrum of Zn-O particles, possibly due to their hexagonal structure [30]. In our study, pristine ZnO spectra (see Appendix A) showed peaks at 122 cm^−1^, 333 cm^−1^, 385 cm^−1^, 438 cm^−1^, 542 cm^−1^, 585 cm^−1^, 1106 cm^−1^ and 1149 cm^−1^.

CHT films containing copper nanoparticles showed additional infrared peaks at 1727 cm^−1^, 1666 cm^−1^, and 1614 cm^−1^ due to C=O functional groups in the copper nanoparticles. When both types of particles are included in the CHT film, the 1537 cm^−1^, 1374 cm^−1^, and the 1093 cm^−1^ absorptions were enhanced. Peaks related to both particles were detected, i.e., 622 cm^−1^ from ZnO and 751 cm^−1^ nCu.

#### 3.1.3. Thermogravimetric Analysis (TGA)

Figure 3 shows TGA and DTGA thermograms of chitosan (CHT) and chitosan mixed with GA (CHT/GA), zinc oxide (CHT/Zn-O), copper nanoparticles (CHT/Cu), and zinc oxide/copper nanoparticles (CHT/Zn-O/Cu). Maximum temperature and 50% weight loss were used to compare thermal stability (Table 2).

Accordingly, with previous studies, degradation occurs between 250–350 °C for unfilled chitosan samples [22]. DTG (Figure 3b) showed that the degradation temperature of the chitosan film is approximately 276 °C (358 °C for 50% mass loss) and a shift at lower temperatures is observed in the modified chitosan films. Ki et al. found two peaks in TGA of chitosan powder, one around 100 °C and the second outstanding at 300 °C; the first corresponding to the vaporization of water contained in the polymer and the second to the pyrolysis of the main chitosan chains and vaporization of the volatile gases produced [21]. In previous studies, weight loss in the first stage has been attributed mainly to water loss, while in the second and third stages it may be due to the breakdown of the main chain and the breakdown of glucose units [25].

When chitosan is crosslinked with glutaraldehyde, a Td is recorded at 262 °C but at 375 °C for loss of 50% of mass (Table 2), which can be explained as a GA crosslinked improve the thermal stability of the chitosan films.

However, Li et al. reported that the thermal stability of chitosan with GA is lower than that of chitosan regardless of temperature [25].

With respect to 5% zinc oxide and 1% copper particles, the decomposition temperature decreases compared to the other samples, which would indicate a lower thermal stability, when these metals are added.

The residual mass of chitosan is significantly lower than the crosslinked chitosan with GA, and this behavior could be explained by the previously mentioned thermal stability improvement of chitosan when crosslinked. Regarding the composite materials, it was observed a similar improvement of thermal stability with the addition of metallic particles; since composites were not prepared with crosslinked chitosan, the expected residual mass was between 11.5% for CHT/Cu, to 16.5% for CHT/Zn-O/Cu, but results showed a higher residual mass percentage after thermal degradation. This unusual thermal decomposition behavior might be attributed to the stability of chitosan by a chelation effect in the presence of metallic particles [36].

#### 3.1.4. X-ray Diffraction (XRD)

Figure 4 shows the diffractograms of pristine chitosan, and chitosan mixed with GA (CHT/GA), zinc oxide (CHT/Zn-O), copper nanoparticles (CHT/Cu) and zinc oxide/copper nanoparticles (CHT/Zn-O/Cu).

Pristine chitosan (Figure 4) has an orthorhombic unit cell with a main peak at 2θ = 9.6°, another at a 2θ = 20.1° and a last one at 2θ = 5.7°. The peak at 2θ = 20° is attributed to crystalline phase I and the angle of greatest intensity to crystalline phase II, which present less hydrated and rigid chains dispersed in an amorphous phase [37]. In addition, pristine chitosan film exhibited greater intensity in the peaks 2θ = 9.6° and 2θ = 20.1° than the films crosslinked with GA and mixed with Cu an Zn-O. This effect could indicate a higher crystallinity in the unmodified films, which reduces with the presence of particles i.e., interfering with crystal formation [38].

Similarly, films crosslinked with GA exhibited a lower intensity peak at 2θ = 19.7°, which coincides with previous studies [8].

X-ray diffraction of CHT/Zn-O films revealed peaks at 2θ = 3.85°, 19.6° and a small peak at 32.44°, also found by Beyene [39] despite the low concentration used for composite preparation. Ai et al. found that Zn-O exhibited peaks at 2θ = 31.3°, 36.2°, 47.5°, and 56.6° that correspond to the crystallographic planes 110, 101, 102, and 110 respectively [40]. The peak found at 2θ = 19.6°, also observed by Youssef et al., is related to the presence of chitosan and of the Zn-O nanoparticles [31]. The diffraction pattern of CHT modified with cooper nanoparticles evidence the presence of Cu_2_O due to the peak found at 2θ = 26.8° corresponding to the 110 crystallographic planes. In a study related to the stabilization of copper and silver nanoparticles in chemically modified chitosan colloidal solution, the patterns exhibited sharp peaks observed at angles at 2θ = 16.9°, 22.5° and 26.8° corresponding to crystallographic planes 110, 210 and 002 [41]. Cu_2_O peaks at approximately at 2θ = 36° have been reported, but these were not observed in our samples [19].

### 3.2. Surface Properties of Modified Chitosan Films

#### 3.2.1. Elemental Composition by EDX and XPS

Elemental composition is shown in Table 3. It is observed by XPS that the percentage of C remains stable in all the films with a slight decrease in the chitosan film crosslinked glutaraldehyde. In contrast, an increase in O was noticed in the same film. The analysis by EDX (Table 3) showed that the C increased from 52.6% to 62% in the GA crosslinked chitosan film while N was not detected. In this regard, Oyrton et al. have reported that the presence of glutaraldehyde causes an increase in the percentage of carbon, and hydrogen and a decrease for nitrogen with the increasing degree of glutaraldehyde [42]. In contrast, Li et al. observed a decrease in C in the chitosan film with glutaraldehyde [25]. Nitrogen, being a low atomic number element was difficult to be detected in a consistent manner in the Cu and Zn-O modified chitosan films. Similarly, Zn-O was not clearly detected by either XPS or EDX in CHT/Zn-O films but XPS showed lower than expected content in chitosan films modified with Zn-O and Cu. Although the presence of Cu in CHT modified Cu films was expected, this was only observed by XPS, possibly because the very low (1%) concentration of Cu added to the films or because it was not present on the surface.

#### 3.2.2. Surface Morphology by SEM

Figure 5 shows the morphology of pristine CHT films, CHT crosslinked with GA, and modified with Zn-O and C. All the films exhibited a smooth surface. However, in chitosan film with Zn-O and copper particles, these particles can be observed on the surface as agglomerates or poorly dispersed particles.

Li et al. observed a rough and porous surface of chitosan crosslinked with glutaraldehyde possibly due to insufficient crosslinking of polymers, or because the glutaraldehyde groups partially grafted on the chitosan, which would indicate that the reaction occurs superficially [25]. It is suggested that the presence of a lower content of copper nanoparticles avoids massive deposition and agglomeration of the particles on the film [17]. Gritsch et al. [35], evaluated films of chitosan and chitosan with Cu. The samples were characterized by having a homogeneous and smooth surface, which coincides with our results.

#### 3.2.3. Contact Angle Measurement

Contact angle measurements were carried out to investigate the hydrophobicity/hidrophillicty of the films. Figure 6 shows the shape of the drop of distilled water (DH_2_O), phosphate buffer (PBS), and culture medium (DMEM) on the modified CHT surface while Table 4 compilates the corresponding contact angle measured.

As shown in Table 4, pristine CHT exhibited similar values of contact angles (88°) independently of the solutions used suggesting a weakly hydrophilic surface as reported by Drelich et al. [43]. Baroudi et al. mention that chitosan has a greater ability to form hydrogen bonds with water molecules, thus destroying intermolecular interactions in the chitosan chains, and when the water droplets are deposited on the chitosan membranes they are immediately absorbed [44].

For GA crosslinked CHT films, there was a slight increase in water contact angle. However, the difference is too small to indicate that the GA film is more hydrophobic. Kang et al. reported similar results due to the relatively hydrophobic GA leading to a slightly decreased surface wettability [45]. In agreement with this, Beppu et al. concluded that crosslinking with GA makes chitosan more hydrophobic, since amino groups are blocked with aliphatic chains [46]. Likewise, Chen et al. observed in chitosan and silicotungstic acid hydrate membranes that the contact angle increased when GA was added, attributed to a decrease in the hydrophilic OH or NH_2_ groups on the membranes because of the reaction between the OH or N–H groups of CHT and the aldehyde groups of GA [26]. The water contact angle of the modified surfaces with Zn-O, Cu, and Zn-O/Cu, showed angles of 89.01°, 88.4°, and 89.51°. respectively, so the addition of metals like copper did not affect their hydrophobic/hydrophilic balance. The same effect was observed for DMEM. The film with higher contact angle (103°) was CHT with Zn-O in DMEM, this may be because adding the zinc oxide particles reduces the hydrophilicity of the film or because de surface roughness is changed. When PBS was used as the sessile drop similar contact angles were observed (87°) in CHT with copper particles. Gritsch et al. confirmed our results as they did not find significant differences in the contact angles of chitosan and its versions modified with copper that were in the range of 75° and 90°. However, changes in the angles of the chitosan samples appeared over time as their valued reduce half their initial value, which does not occur in the samples with copper and this behavior is attributed to the chelation of the same ion. Adjacent chitosan chains induce a crosslinking of the polysaccharide matrix that is able to reduce cracks on the surface of the sample [35]. The contact angle is known to be greater for rougher surfaces than for smoother surfaces [47]. It is important to mention that the contact angles observed in this study were taken from the air-dried surface, meaning a rough surface, so it could influence the results.

### 3.3. Antimicrobial Activity

Table 5 shows the antibacterial activity of CHT, GA crosslinked CHT, and CHT modified with Zn-O and C films against *Staphylococcus aureus* and *S. typhimurium.*

For pristine chitosan films no antimicrobial activity was found against *Staphylococcus aureus*. Oppositely, Imani et al. reported an adequate antibacterial efficacy of chitosan against *S. aureus* as a medicine for pulpectomy of primary teeth [48]. In contrast, in the pure chitosan films, antimicrobial activity was found against *S. Typhimurium*. These findings are consistent with those reported by Hu et al., who used pure chitosan to reduces the counts of bacteria such as *Salmonella* [49]. Erickson et al. found similar results of a chitosan and acetic acid spray against the same microorganisms [50].

GA crosslinked sample showed antimicrobial activity against *S. aureus* and *S. typhimurium*, this can be attributed to the interaction between positively charged CHT-GA molecules and negatively charged bacterial membranes [51]. 

It was suggested that residual aldehyde of CHT–GA crosslinked films have an cytotoxic bacterial effect, since unreacted terminal aldehyde groups are able to interact with functional groups on the bacterial membrane [52].

Sehmi et al. mention that glutaraldehyde is not active against bacterial cells when it is found in acidic aqueous solutions, however, when it is activated at pH 7.5–8.5 the solution becomes biocidal [53]. The antibacterial activity of CHT/GA film may be due to its high hydrophobicity [25]. The antimicrobial activity of the Zn-O chitosan film was not found against *S. aureus*, but it was observed against *S. typhimurium*. Chekne et al. evaluated the antibacterial effect of the bound gamma globulin fraction and the free copper and zinc cations applied to *S. aureus* cultures. Free copper cations as well as copper bound by serum γ-globulin resulted in a complete bactericidal effect after 4 h of observation. On the other hand, zinc cations perform bacteriostatic activity only bound to γ-globulin; therefore, free zinc cations did not have a bacteriostatic effect against *S. aureus* [54]. Valencia et al. determined the antimicrobial effect of commercial chitosan in liquid and solid state against *Staphylococcus aureus*. Chitosan did not show an antimicrobial effect against S. aureus [12]. Brathi et al. reported that chitosan and Zn-O nanocomposite showed greater antimicrobial activity against Gram-negative bacterial pathogens compared to Gram-positive ones [55]. In contrast, Youssef et al. found antimicrobial activity of ZnO-chitosan films against both bacteria, *S. aureus* and *S. typhimurium* [31].

For copper chitosan film, no antimicrobial activity was observed against *S. aureus* but there was activity against *S. typhimurium*. Our results differed from those reported by Brahma et al., who evaluated a copper complex against *S. aureus*, and found antimicrobial activity [56]. Li et al. mentioned that when Cu concentrations are below a critical value, they would not show antibacterial properties because bacteria are good at controlling copper levels, which could explain the results of our study. Instead, if the critical value of Cu is exceeded the antibacterial activity would improve [57]. Isobe et al. also states that Gram-positive bacteria have a peptidoglycan layer (50–60 mm) under the cell membrane and this layer is known to help bacteria overcome physical stress, so it is believed that this layer of cell wall reduces the penetration of toxic metal ions [58].

It has been reported in several studies that Gram-negative bacteria are more sensitive to chitosan loaded with metal ions probably due to the different characteristics of the cell surfaces, i.e., the negative charge on the cell surface of Gram-negative bacteria was higher than in Gram-positive bacteria [59]. Chung et al. also concluded that Gram-negative bacteria were more susceptible than Gram-positive. This outcome was caused by the greater hydrophilicity of Gram-negative bacteria, making them more susceptible to chitosan in solution [60]. However, in our study, zinc oxide and copper modified chitosan film exhibited *S. aureus* growth.

In contrast, Yasuyuki et al. found that the metals Cu and Zn showed antibacterial properties against Gram-positive bacteria such as *S. aureus*. The studies provided evidence of metal toxicity and antibacterial activity [61]. It has been reported that the minimum bactericidal concentration for *S. aureus* is 16 µg/mL for ZnO while for CuO is 22 µg/mL. [62]. In our study, it is clear that even when a higher amount of ZnO was used compared to Cu nanoparticles, the effective amount released from CHT was not known. Furthermore, agar diffusion methods were originally designed for soluble antibiotics and in this case, solubility and ion release is limited. Another explanation about the low antimicrobial activity of ZnO could be due that we use micrometric and non-nanometric particles.

For *S. typhimurium*, antimicrobial activity was found in chitosan modified with zinc and copper oxide. These results coincide with some studies, where Cu copper has been used as a potential antimicrobial agent against *S. typhimurium* [63,64]. Moreover, Karbowniczek et al. found antimicrobial activity of Zn-O nanoparticles against this microorganism [65].

Regarding the safety of the use of copper and ZnO, several studies provide evidence about cytocompatibility, such as the use of copper nanoparticles in human periodontal ligament stem cells [66]. Another study revealed cytocompatibility and osteogenesis activity of the Ti-Cu in vitro. In addition, Ti-Cu alloy could significantly promote the osteogenic differentiation of MG63 cells by upregulating the osteogenesis-related gene expressions including alkaline phosphatase (ALP), collagen I (Colla I), osteopontin (OPN), and osteocalcin (OCN) [67]. Cell viability of Zn was tested in EA.hy926 cells, reporting an increase with the addition of Cu or Cu and Fe content. The authors report that there was not adversely affected on platelets adhering to the surface of the Zn alloys [68]. According to Zhu, Zn exposure was found to enhance MSC growth and differentiation [69]. 

## 4. Conclusions

Chitosan can be modified with zinc oxide and copper nanoparticles. Despite the low concentrations used in our study for this biomedical application, it can be detected by subtle changes by using FTIR, Raman, TGA, XRD, XPS, and EDX techniques. The incorporation of either ZnO or copper particles has no effect on surface hydrophilicity as a contact angle close to 88° was observed. Therefore, the biological effect will be attributed to their composition only. No antimicrobial activity against *Staphylococcus aureus* was found on unfilled chitosan, chitosan film cross-linked with glutaraldehyde, or on the films with 5% zinc oxide, 1% copper nanoparticles or their combination. The lack of antimicrobial activity of copper and zinc oxide is attributed to the low amount added to the films. In contrast, antimicrobial activity against *Salmonella* was found in all films, i.e., pristine chitosan and chitosan modified with 5% zinc oxide and 1% copper. However, no synergistic effect was observed at the concentrations studied. It is possible that these traditional bactericidal particles (zinc oxide or copper) where not available at the correct oxidation state even when they were observed on the surface by SEM. It is also recommended that longer extraction times are used in order to assess their bactericidal activity. Therefore, it is concluded that chitosan does have an antimicrobial effect at 5% ZnO and 1% NCu for *S. typhimurium* and that is recommended that higher concentrations are used against *Staphylococcus aureus* for medical and dental applications. Further studies are recommended for medical and dental applications.

## Figures and Tables

**Figure 1 polymers-13-03861-f001:**
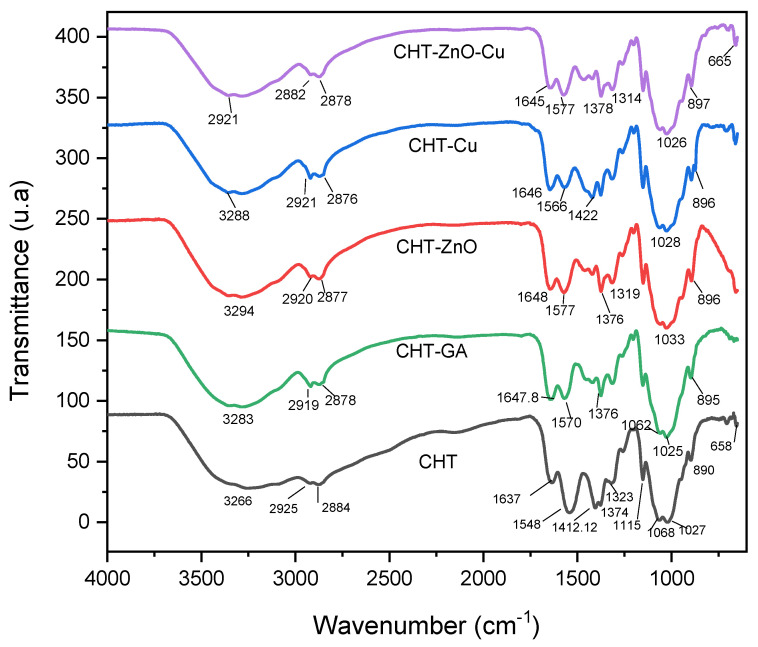
FTIR spectra of CHT, CHT/GA, CHT/Zn-O, CHT/Cu, CHT/Zn-O/Cu.

**Figure 2 polymers-13-03861-f002:**
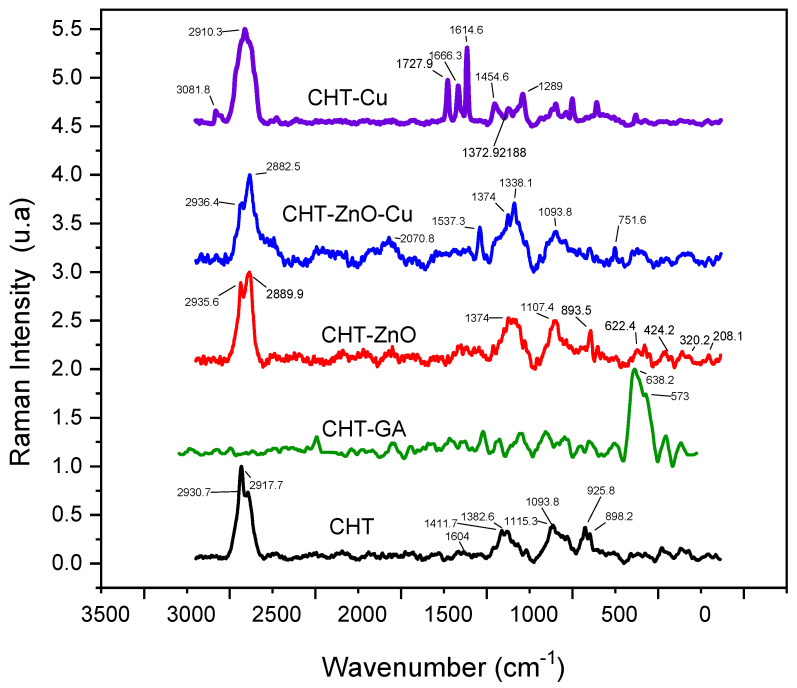
Raman spectra of chitosan crosslinked GA (CHT/GA) chitosan with zinc oxide (CHT/Zn-O), chitosan with copper nanoparticles (CHT/Cu) and chitosan with zinc oxide/copper nanoparticles (CHT/Zn-O/Cu).

**Figure 3 polymers-13-03861-f003:**
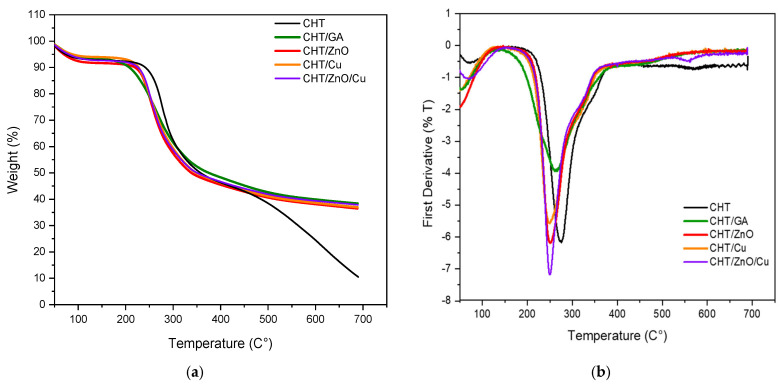
TGA (**a**) and DTGA (**b**) thermograms of chitosan crosslinked GA (CHT/GA) chitosan with zinc oxide (CHT/Zn-O), chitosan with copper nanoparticles (CHT/Cu), and chitosan with zinc oxide/copper nanoparticles (CHT/Zn-O/Cu).

**Figure 4 polymers-13-03861-f004:**
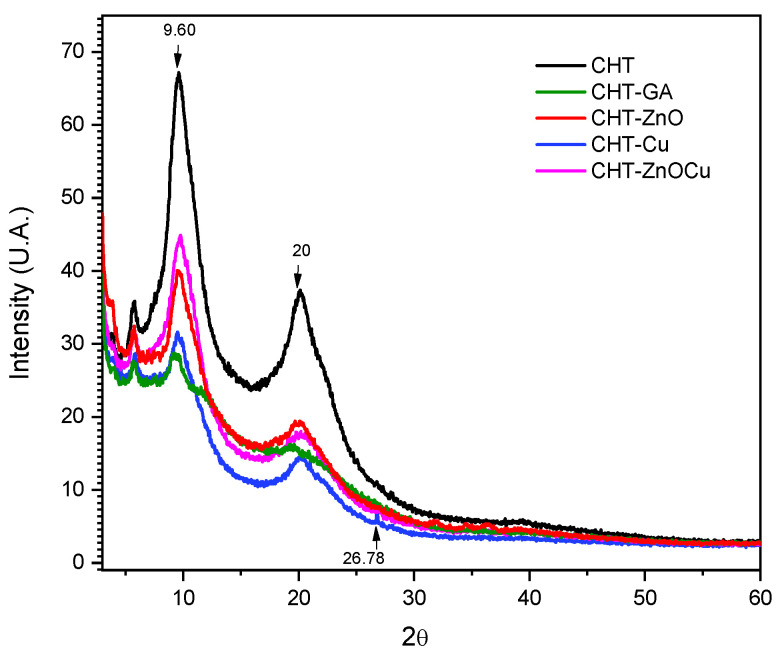
XRD pattern of CHT chitosan crosslinked GA (CHT/GA) chitosan with zinc oxide (CHT/Zn-O), chitosan with copper nanoparticles (CHT/Cu), and chitosan with zinc oxide/copper nanoparticles (CHT/Zn-O/Cu).

**Figure 5 polymers-13-03861-f005:**
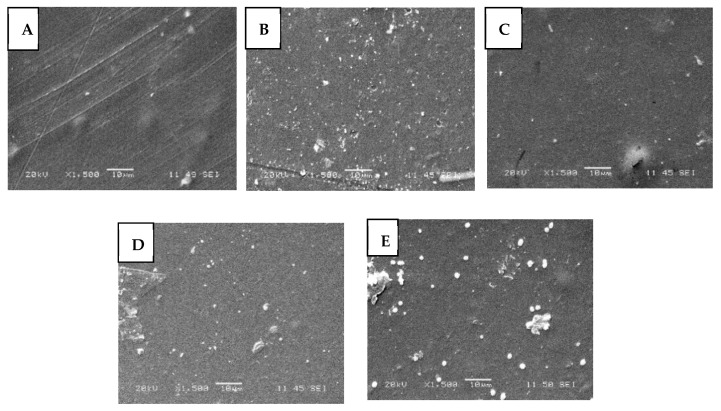
SEM surface morphology of (**A**) pristine CHT films, (**B**) CHT crosslinked GA, (**C**) CHT with Zn-O, (**D**) CHT with Cu, and (**E**) CHT with Zn-O and Cu.

**Figure 6 polymers-13-03861-f006:**
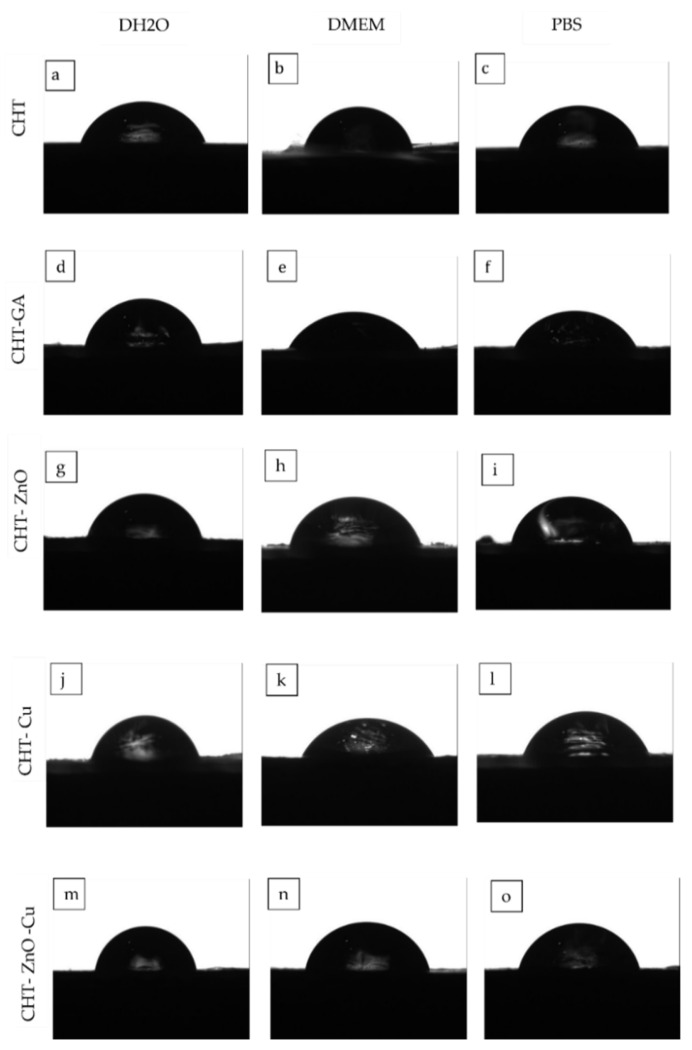
Contact angles of CHT (**a**–**c**), CHT crosslinked GA (**d**–**f**), CHT with Zn-O (**g**–**i**), CHT with Cu (**j**–**l**), CTH with Zn-O and Cu (**m**–**o**).

**Table 1 polymers-13-03861-t001:** Composition of chitosan and chitosan composite films.

	CHT	CHT + GA	CHT + ZnO	CHT + Cu	CHT+ ZnO + Cu
Chitosan	200 mg	200 mg	190 mg	198 mg	188 mg
Glutaraldehyde		150 µL			
ZnO 5 wt.%			10 mg		10 mg
Copper 1 wt.%				2 mg	2 mg

**Table 2 polymers-13-03861-t002:** Decomposition temperatures and weight loss and residual mass at 700 °C of CHT films, CHT/GA, CHT/Zn-O, CHT/Cu and CHT/Zn-O/Cu.

Films	T_d_ (°C)	T (°C) at 50% of Weight Loss	Residual Mass (%)
CHT	276	358	10.45
CHT/GA	262	375	38.38
CHT/Zn-O	251	338	36.47
CHT/Cu	247	348	36.96
CHT/Zn-O/Cu	249	350	37.89

**Table 3 polymers-13-03861-t003:** XPS and EDX atomic percentage of C, O, and N of chitosan films crosslinked with GA and mixed with Zn-O and Cu.

	CHT		CHT/GA		CHT/ZnO		CHT/Cu		CHT/ZnO/Cu	
%	XPS	EDX	XPS	EDX	XPS	EDX	XPS	EDX	XPS	EDX
C	88.59	52.64	77.52	62	87.28	64.69	85.6	58.62	87.93	55.13
O	11.41	41.73	18.63	38	11.09	33.54	12.63	38.82	11.4	41.73
N		5.63	3.85		1.63	1.72	1.77	2.5		3.12
Zn						0.05			0.66	
Cu								0.07		0.02

**Table 4 polymers-13-03861-t004:** Contact angle of the pristine CHT and CHT modified films.

Film	H_2_O (°)	DMEM (°)	PBS (°)
CHT	88.29 ± 0.33	88.60 ± 0.54	88.41 ± 0.33
CHT-GA	88.75 ± 0.22	87.35 ± 0.56	87.31 ± 1.14
CHT-ZnO	89.0 ± 0.39	103.33 ± 22.72	89.00 ± 0.53
CHT-Cu	88.40 ± 0.58	87.16 ± 0.49	87.98 ± 0.83
CHT-ZnO-Cu	89.51 ± 0.41	88.41 ± 0.69	89.01 ± 0.35

**Table 5 polymers-13-03861-t005:** Antimicrobial activity: positive (+) and negative (−) against *Staphylococcus aureus* and *S. tiphimurium* of chitosan discs; before 24 h (top) and after 24 h (bottom) of inoculation.

Film	Antibacterial Activity *Staphylococcus Aureus*	Antibacterial Activity*Salmonella Typhimurium*
CHT	(−) 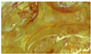 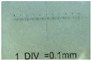	(+) 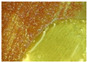 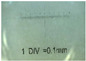
CHT + GA	(+) 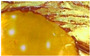 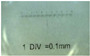	(+) 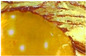 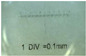
CHT + Zn-O	(−) 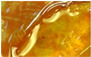 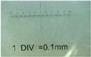	(+) 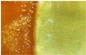 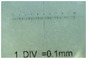
CHT + Cu	(−) 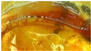 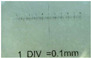	(+) 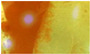 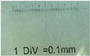
CHT+ ZnOCu	(−) 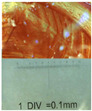	(+) 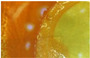 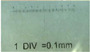

## Data Availability

The data presented in this study are available on request from the corresponding author. The data are not publicly available due to its content is part of a Master´s Thesis.

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
