# Peer review of "Zinc Oxide and Copper Chitosan Composite Films with Antimicrobial Activity"

_polymers, 2021, doi:10.3390/polym13223861_

Round 1
Reviewer 1 Report
The manuscript titled “Zinc oxide and copper chitosan composite films with antimicrobial activity” describes fabrication, characterization and application of nanocomposite based on chitosan, ZnO and CuNPs for antibacterial applications. It possesses some merits for applications and can be considered to publish in Polymers after the major corrections. The authors should follow the below comments:
- Many researches have reported synthesis of ZnO/Chitosan and Cu/Chitosan for multiple applications. The authors must point out novelty of this work compared to the previous reports.
- In the preparation of composite films, the authors did not use GA for making crosslinks, however, they did a comparable study between samples and GA-chitosan. Please explain this study route.
- The size and morphology of ZnO and Cu nanoparticles as well as their mixture in chitosan must be determined via TEM technology.
- For TGA spectra, the ash content of samples remained after 700 oC should be added to Table 2. The ash content of all composite including GA-Chitosan, ZnO-Chitosan, Cu-Chitosan and ZnO-Cu-Chitosan is not almost different although content of ZnO was used up to 5% w/w. However, this ash content is a big difference from thermal property of chitosan. The authors must discuss this point.
- Please check typos and italic style of bacterium names thoroughly.
Author Response
Response to Reviewer 1 Comments
Thank you very much for reviewing our manuscript. We greatly appreciate your comments and suggestions. We have revised the manuscript accordingly. Please find attached a point-by-point response to reviewer’s concerns. We hope that you find our responses satisfactory and that the manuscript is now acceptable for publication.
Point 1: Many researches have reported synthesis of ZnO/Chitosan and Cu/Chitosan for multiple applications. The authors must point out novelty of this work compared to the previous reports.
Response 1: Appearance of microorganisms resistant to antibiotics is commonly found in the oral cavity. Chitosan has outstanding properties that are suitable for usage in oral tissue; hence, as a response to our search for improved antimicrobial dental materials, we selected chitosan as a matrix for our study. To assess the synergic effect of this matrix with other antibacterial agents, this research sought to simultaneously integrate zinc oxide particles and low concentration copper nanoparticles. Although previous studies have demonstrated that ZnO nanoparticles has excellent antibacterial activity against both Gram-positive and Gram-negative bacteria [Azam A, Ahmed AS, Oves M, Khan MS, Habib SS, Memic A. Antimicrobial activity of metal oxide nanoparticles against Gram-positive and Gram-negative bacteria: a comparative study. Int J Nanomedicine. 2012;7:6003-9.], it seems that this behavior is enhanced when used ZnO as nanoparticles which was not the case in our study. The novelty of this research is not only that it provides evidence of the synergy of the mentioned components – ZnO, Cu and chitosan – in their antimicrobial behavior, but it also provides evidence of the structural changes induced in the matrix, that has not been fully addressed by previous studies. Furthermore, glutaraldehyde crosslinked chitosan, sometimes not used for their toxicity, was used as a third alternative. The foregoing focused on being used for diseases that affect the oral cavity, such as dental caries. These additives (chitosan, glutaraldehyde, ZnO and Cu) do not generate bacterial resistance and no secondary allergic reactions have been reported.
To date, there are no reports of these two compounds (ZnO and Cu) in conjunction with chitosan for use in the oral cavity, thus opening the field for future research.
Also, we modified the manuscript adding the following texts:
Finally, we modified the original manuscript by adding the following text
Lines 18 – 32:
“The role of the oral microbiome and its effect on dental diseases is gaining interest. Therefore, it has been sought to decrease the bacterial load to fight oral cavity diseases. In this study, composite materials based on chitosan, chitosan crosslinked with glutaraldehyde, chitosan with zinc oxide particles, and chitosan with copper nanoparticles were prepared in the form of thin films, to evaluate a new alternative with a more significant impact on the oral cavity bacteria. The chemical structures and physical properties of the films were characterized using by Fourier transform infrared spectroscopy (FTIR,) Raman spectroscopy, X-ray photoelectron spectroscopy (XPS), elemental analysis (EDX), thermogravimetric analysis (TGA), X-ray diffraction (XRD), scanning electron microscopy (SEM) and contact angle measurements. Subsequently, the antimicrobial activity of each material was evaluated by agar diffusion tests. No differences were found in the hydrophilicity of the films with the incorporation of ZnO or copper particles. Antimicrobial activity was found against S. aureus in the chitosan film crosslinked with glutaraldehyde, but not in the other compositions. In contrast antimicrobial activity against S. typhimurium was found in all films. Based on the data of present investigation, chitosan composite films could be an option for the control of microorganisms with potential applications in various fields, such as medical and food industry.”
Lines 78 – 81
“At the same time, to provide some evidence about the synergy of the components, ZnO, Cu and chitosan, arranged in the same material, and determining if their antimicrobial capacity improves when used together.”
Point 2: In the preparation of composite films, the authors did not use GA for making crosslinks, however, they did a comparable study between samples and GA-chitosan. Please explain this study route.
Response 2: Glutaraldehyde crosslinked chitosan, sometimes not used for their toxicity, was used as a third alternative as it is an effective crosslinker for CHT, and also it is commonly employed as a high-level disinfectant for dental equipment. However, their effect on materials for dental applications deserve further investigation due to their high mineralization potential.
To clarify the methodology of chitosan films crosslinked GA, the following text have been added to the manuscript, in lines 102- 109:
“2.2.2. Preparation of chitosan composite films
Preparation of glutaraldehyde crosslinked chitosan films
Crosslinked GA- chitosan was obtained dissolved 200 mg of chitosan in 30 mL of acetic acid. The solution was stirred in a magnetic stirrer plate at 25 °C, for one hour. After the dissolution of chitosan, 0.3744 mM (150 µL) of a 25% glutaraldehyde (GA) was added. The mixture was left under magnetic stirring for 5 hours until complete homogenization of the mixture and then the solution was poured into plastic Petri dishes and dried at 25 °C. Films obtained after solvent casting were neutralized with 5 wt.% NaOH and rinsed with distilled water. Finally, films dried at 25° for 2 days.”
On the other hand, we decided use GA crosslinked chitosan films, as a control of the antimicrobial activity, according to reported in a previous study. (Appl. Sci. 2021, 11(3), 1267; https://doi.org/10.3390/app11031267).
Also, we provided more information related to GA antibacterial effect:
Lines 473- 479
“It was suggested that residual aldehyde of CHT-GA crosslinked films have an cytotoxic bacterial effect, since unreacted terminal aldehyde groups are able to interact with functional groups on the bacterial membrane”
Point 3: The size and morphology of ZnO and Cu nanoparticles as well as their mixture in chitosan must be determined via TEM technology.
Response 3: We authors, respectfully consider, that filling particles (ZnO and Cu) are commercial products described by manufacturers, including size, morphology, and other relevant characteristics. Regarding particles mixture in chitosan, TEM technology is an accurate analysis which determines distribution of nanoparticles in composite materials; nevertheless, there are other characteristics associated with the mixture of particles with chitosan that are reported, explained, and discussed in our manuscript. The text refers to this matter in the sections of XRD analysis, EDX and XPS analyses, and specifically in the surface morphology analysis by SEM section.
The text of section “2.1. Materials” (lines 88-92) was modified, by adding the ZnO product CAS and average particle size.
“Chitosan (molecular weight 223.332 g/mol and with 70-80% deacetylation degree), Acetic Acid, Sodium Hydroxide, Glutaraldehyde grade II at 25% (molar mass 100.11 g/mol) and zinc oxide (ZnO) (powder, ASC reagent, CAS 1314-13-2, SKU 205532) reagents were acquired from Sigma-Aldrich. Copper nanoparticles (nCu) in aqueous suspension were acquired from the manufacturer Nano Process SPA. The determination of nCu particle size was performed by dynamic light scattering (DLS) using a NANOTRAC WAVE II, following the methodology described elsewhere [19]; the average particle size was 27.4 nm. The particle size of ZnO was 55 μm, measured by LS100 Coulter Particle Size Analyzer.
Point 4: For TGA spectra, the ash content of samples remained after 700 oC should be added to Table 2. The ash content of all composite including GA-Chitosan, ZnO-Chitosan, Cu-Chitosan and ZnO-Cu-Chitosan is not almost different although content of ZnO was used up to 5% w/w. However, this ash content is a big difference from thermal property of chitosan. The authors must discuss this point.
Response 4: The residual mass of materials after thermal degradation at 700 ºC is now reported on table 2. Additionally, a discussion regarding these findings has been added to text (in lines 308 -320), Including further supporting reference. The new text reads as follows:
“The residual mass of chitosan is significantly lower than the crosslinked chitosan with GA, this behavior could be explained by the previously mentioned thermal stability improvement of chitosan when crosslinked. Regarding the composite materials, it was observed a similar improvement of thermal stability with the addition of metallic particles; since composites were not prepared with crosslinked chitosan, the expected residual mass was between 11.5 % for CHT/Cu, to 16.5 % for CHT/Zn-O/Cu, but results showed a higher residual mass percentage after thermal degradation. This unusual thermal decomposition behavior might be attributed to the stability of chitosan by a chelation effect in the presence of metallic particles. (Wei Li, Ling Xiao & Caiqin Qin (2010): The Characterization and Thermal Investigation of Chitosan-Fe3O4 Nanoparticles Synthesized Via a Novel One-step Modifying Process, Journal of Macromolecular Science, Part A: Pure and Applied Chemistry, 48:1, 57-64).
Table 2. Decomposition temperatures, weight loss and residual mass at 700°C of CHT films, CHT/GA, CHT/Zn-O, CHT/Cu and CHT/Zn-O/Cu.
|
Films |
Td (°C) |
T (°C) at 50% of weight loss |
Residual mass (%) |
|
CHT |
276 |
358 |
10.45 |
|
CHT/GA |
262 |
375 |
38.38 |
|
CHT/Zn-O |
251 |
338 |
36.47 |
|
CHT/Cu |
247 |
348 |
36.96 |
|
CHT/Zn-O/Cu |
249 |
350 |
37.89 |
Point 5: Please check typos and italic style of bacterium names thoroughly.
Response 5: Thank you for such an important observation. We appreciate your thoroughly readding, and accordingly, we checked and resolved typos and italic style of bacterium names.

Reviewer 2 Report
The manuscript is interesting and worth of publication, however I recommend the major revision befor publication, as follows:
Abstract should be rewritten. Sentences from Lines 18-23 are rather for Introductions. Abstract should be more compact and focused on the obtained results. Last sentence should be in the end of conlusions
Glutaraldehyde addition is not clear, please clarify.
The addition of ZnO nanoparticles is not clear. It was a water solution or powder of ZnO?
Line 175 abbreviations should be omitted in titles of sections
Table 4 „°” should be only in the top, not next to all results
Please check the correctness of citations
Some safety aspects should be taken into account in discussion
Practical applications should be more specify, i.e. dose
Author Response
Response to Reviewer 2 Comments
Dear Reviewer 2
Thank you very much for reviewing our manuscript. We greatly appreciate your comments and suggestions. We have revised the manuscript accordingly. Please find attached a point-by-point response to reviewer’s concerns. We hope that you find our responses satisfactory and that the manuscript is now acceptable for publication.
Point 1: Abstract should be rewritten. Sentences from Lines 18-23 are rather for Introductions. Abstract should be more compact and focused on the obtained results. Last sentence should be in the end of conclusions.
Response 1: Considering your suggestion, the abstract, in lines 18 – 32, was rewritten as follows:
“The role of the oral microbiome and its effect on dental diseases is gaining interest. Therefore, it has been sought to decrease the bacterial load to fight oral cavity diseases. In this study, composite materials based on chitosan, chitosan crosslinked with glutaraldehyde, chitosan with zinc oxide particles, and chitosan with copper nanoparticles were prepared in the form of thin films, to evaluate a new alternative with a more significant impact on the oral cavity bacteria. The chemical structures and physical properties of the films were characterized using by Fourier transform infrared spectroscopy (FTIR,) Raman spectroscopy, X-ray photoelectron spectroscopy (XPS), elemental analysis (EDX), thermogravimetric analysis (TGA), X-ray diffraction (XRD), scanning electron microscopy (SEM) and contact angle measurements. Subsequently, the antimicrobial activity of each material was evaluated by agar diffusion tests. No differences were found in the hydrophilicity of the films with the incorporation of ZnO or copper particles. Antimicrobial activity was found against S. aureus in the chitosan film crosslinked with glutaraldehyde, but not in the other compositions. In contrast antimicrobial activity against S. typhimurium was found in all films. Based on the data of present investigation, chitosan composite films could be an option for the control of microorganisms with potential applications in various fields, such as medical and food industry.”
Point 2: Glutaraldehyde addition is not clear, please clarify.
Response 2: In accordance with your suggestion, we complete the information on section 2.2.2 lines 102 – 109:
“2.2.2. Preparation of glutaraldehyde crosslinked chitosan films
Crosslinked GA- chitosan films was obtained dissolved 200 mg of chitosan in 30 mL of acetic acid. The solution was stirred in a magnetic stirrer plate at 25 °C, for one hour. After the dissolution of chitosan, 0.3744 mM (150 µL) of a 25% glutaraldehyde (GA) was added. The mixture was left under magnetic stirring for 5 hours until complete homogenization of the mixture and then the solution was poured into plastic Petri dishes and dried at 25 °C. Films obtained after solvent casting were neutralized with 5 wt.% NaOH and rinsed with distilled water. Finally, films dried at 25° for 2 days”
Point 3: The addition of ZnO nanoparticles is not clear. It was a water solution or powder of ZnO?
Response 3: We performed our experiment with powder of ZnO
The following texts have been added to the manuscript:
Line 86: “powder, ASC reagent, CAS 1314-13-2, SKU 205532”
Point 4: Line 175 abbreviations should be omitted in titles of sections
Response 4:
The title of section 3.1 (line 184): was changed as follows:
“3.1. Physicochemical and structural characterization of modified CHT films”
Point 5: Table 4 “°” should be only in the top, not next to all results
Response 5:
All the “°” was removed from the content of the Table 4:
Table 4. Contact angle of the pristine CHT and CHT modified films.
|
Film |
H2O (°) |
DMEM (°) |
PBS (°) |
|
CHT |
88.29 ± 0.33 |
88.60 ± 0.54 |
88.41 ± 0.33 |
|
CHT-GA |
88.75 ± 0.22 |
87.35 ± 0.56 |
87.31 ± 1.14 |
|
CHT-ZnO |
89.0 ± 0.39 |
103.33 ± 22.72 |
89.00 ± 0.53 |
|
CHT-Cu |
88.40 ± 0.58 |
87.16 ± 0.49 |
87.98 ± 0.83 |
|
CHT-ZnO-Cu |
89.51 ± 0.41 |
88.41 ± 0.69 |
89.01 ± 0.35 |
Point 6: Please check the correctness of citations
Response 6: Thank you for the observation. We revied all the citations in detail, and corrected the typos and errors in accordance with the requested style of citation.
Point 7: Some safety aspects should be taken into account in discussion
Response 7: The discussion about safety and cell viability has been expanded as follows:
Lines 489 – 498
“Regarding the safety of the use of copper and ZnO, several studies provide evidence about cytocompatibility, such as the use of copper nanoparticles in human periodontal ligament stem cells [1]. Another study revealed cytocompatibility and osteogenesis activity of the Ti-Cu in vitro. In addition, Ti-Cu alloy could significantly promote the osteogenic differentiation of MG63 cells by upregulating the osteogenesis-related gene expressions including alkaline phosphatase (ALP), Collagen I (Colla I), osteopontin (OPN) and osteocalcin (OCN) [2]. Cell viability of Zn was tested in EA.hy926 cells, reporting an increase with the addition of Cu or Cu and Fe content. The authors report that there was not adversely affected on platelets adhering to the surface of the Zn alloys [3]. According to Zhu, Zn exposure was found to enhance MSC growth and differentiation “
Point 8: Practical applications should be more specify, i.e. dose
Response 8: Although toothbrushing is considered sufficient to fight pathogenic microorganisms of the oral cavity, we consider of utmost importance to find novel alternatives aimed at improving dental materials, for them to have a more significant impact on the oral cavity bacteria. One alternative is the usage of modified antibiotics biomaterials which are defined as any material that inhibit or prevent the multiplication of bacteria, fungi, and parasites Ironically, antimicrobial resistance has increased in parallel with its overused; hence, we are invested in finding materials that have bactericidal properties and, at the same time, will not generate antibacterial resistance. Antimicrobial properties of chitosan, ZnO and Cu are proved against species of bacteria commonly responsible for the main oral diseases such as dental caries and periodontal disease. The material that we propose based on chitosan with copper and ZnO might be effective for the control of dental biofilm. Furthermore, it could be used as a liner to avoid the reinfection of microorganisms. It could also be used as a coating for dental prosthesis to prevent subprosthetic stomatitis, orthodontic appliances, and mouthguards to avoid inflammation and mucositis of the oral cavity soft tissues. The dose should be sufficient to be bactericidal without compromising cell viability. This will depend on the type of particle added. It is accepted that gram negative bacteria require higher doses than gram positive bacteria. However, more in vitro, and in vivo studies are needed to be able to determine the optimal dose that possesses the desirable antimicrobial properties without causing toxicity.
Reference about the doses reported was incorporated into the manuscript. Accordingly, the following text was added to results and discussion section:
Lines 479 - 484
“It has been reported that the minimum bactericidal concentration for S. aureus is 16 µg/mL for ZnO while for CuO is 22 µg/mL. [Azam A, Ahmed AS, Oves M, Khan MS, Habib SS, Memic A. Antimicrobial activity of metal oxide nanoparticles against Gram-positive and Gram-negative bacteria: a comparative study. Int J Nanomedicine. 2012;7:6003-9.] In our study, it is clear that even when a higher amount of ZnO is used compared to Cu nanoparticles, the effective amount released from CHT is not known. Furthermore, agar diffusion methods were originally designed for soluble antibiotics and in this case, solubility and ion release is limited. Another explanation about the low antimicrobial activity of ZnO could be due that we use micrometric and non-nanometric particles”

Round 2
Reviewer 1 Report
The reviewer recognizes the affords for improvement of their manuscript. They responded and corrected most of comments. However, the authors refused to provide TEM images which can confirm the morphology and nanoparticle distribution on the film. They claimed that “there are other characteristics associated with the mixture of particles with chitosan that are reported, explained, and discussed in our manuscript.”. Unfortunately, these characterizations (TEM, XRD, XPS) are impossible to confirm distribution and morphology in nano-scale. It should be noted that the manuscript used Cu nanoparticles which is major agent for improvement of bioactivity in the film but it is not demonstrated clearly. Therefore, I cannot recommend to publish this work.
However, if the other reviewers accept the manuscript, editors can make a decision.
Author Response
Thank you for your comments
Reviewer 2 Report
Tha authors took into account all comments and suggestions. The manuscript was corrected and can be published in the present formAuthor Response
Thank you very much for your comments.